# Wide Field of View Under-Panel Optical Lens Design for Fingerprint Recognition of Smartphone

**DOI:** 10.3390/mi15030386

**Published:** 2024-03-13

**Authors:** Cheng-Mu Tsai, Sung-Jr Wu, Yi-Chin Fang, Pin Han

**Affiliations:** 1Graduate Institute of Precision Engineering, National Chung Hsing University, Taichung City 402, Taiwan; jmutsai@email.nchu.edu.tw (C.-M.T.); tiger.wuss@smail.nchu.edu.tw (S.-J.W.); 2Institute of Smart Industry and Green Energy, National Yang Ming Chiao Tung University, Tainan City 711, Taiwan

**Keywords:** LED-backlight, fingerprint recognition under the screen, optical design, tolerance analysis

## Abstract

Fingerprint recognition is a widely used biometric authentication method in LED-backlight smartphones. Due to the increasing demand for full-screen smartphones, under-display fingerprint recognition has become a popular trend. In this paper, we propose a design of an optical fingerprint recognition lens for under-display smartphones. The lens is composed of three plastic aspheric lenses, with an effective focal length (EFL) of 0.61 mm, a field of view (FOV) of 126°, and a total track length (TTL) of 2.54 mm. The image quality of the lens meets the target specifications, with MTF over 80% in the center FOV and over 70% in the 0.7 FOV, distortion less than 8% at an image height of 1.0 mm, and relative illumination (RI) greater than 25% at an image height of 1.0 mm. The lens also meets the current industry standards in terms of tolerance sensitivity and Monte Carlo analysis.

## 1. Introduction

Fingerprint recognition [1,2] is a common biometric recognition technology that has the advantages of being fast, convenient, and secure. It is widely used in smartphones, laptops, access control systems, and other fields. The effectiveness of this process hinges on the quality of the captured fingerprint image, which is largely determined by the performance of the fingerprint lens. Its ability to accurately focus light and gather sufficient detail from the complex fingerprint ridges makes it a crucial component of the entire recognition system. 

Several technologies have been proposed for capturing high-fidelity fingerprint images, including micro-lens arrays (MLAs) [3,4,5], holographic lenses [6,7,8], and wide-angle lenses [9]. MLAs utilize an array of miniature lenses to precisely focus fingerprint details onto a sensor. This allows for a remarkably compact sensor design, ideal for space-constrained applications like mobile devices. Holographic lenses leverage the power of holography to achieve exceptional image quality at a remarkably small size. Their ability to deliver high resolution and contrast translates to accurate fingerprint capture. Both MLAs and holographic lenses currently struggle to capture large field-of-view (FOV) fingerprint images. This limitation restricts their suitability for applications requiring wider area coverage. 

Wide-angle lenses [10,11] offer a promising approach for capturing an expansive FOV in fingerprint recognition systems. However, their design often introduces significant distortion and reduces peripheral relative illumination. Digital image processing techniques can partially address distortion through software-based methods [12]. Notably, while Sun et al. [11] proposed a wide-angle lens design mitigating low peripheral relative illumination, wide-angle lenses specifically designed for fingerprint recognition require a shorter lens length. The primary challenge lies in minimizing the total track length (TTL) while maintaining optimal image quality. Lai [9] presented a wide-angle lens with a 116° FOV and a compact total length of 2.44 mm, addressing the TTL concern for fingerprint applications. However, its image recognition performance remains suboptimal. To overcome this limitation, we propose a wider FOV lens design with a 126° FOV, and the total length is almost the same (2.54 mm). This design achieves a minimum relative illumination (RI) exceeding 25% at the image edges. Such a lens design ensures a modulation transfer function (MTF) exceeding 80% in the center FOV and over 70% in the 0.7 FOV, guaranteeing high-fidelity fingerprint capture. 

Furthermore, rigorous tolerance analysis [13,14,15] confirms that the proposed lens can be manufactured with a 98% yield within existing production tolerances, paving the way for its large-scale deployment in fingerprint recognition systems. The remaining sections of this paper are structured as follows. Section 2 details the pre-design considerations for the under-panel lens, grounded in Gaussian optics principles. Section 3 then elaborates on the lens optimization process and presents the associated simulation outcomes. Subsequently, Section 4 implements a tolerance analysis to assess the feasibility of mass production. Finally, Section 5 draws concise conclusions from the presented work. 

## 2. Pre-Design of Under-Panel Lens Based on Gaussian Optics 

An image lens system, as depicted in Figure 1, has a known object distance *l* and an effective focal length (EFL) *F*. The conjugate equation based on Gaussian optics [16]
(1)$$\frac{1}{l^{\prime }}-\frac{1}{l}=\frac{1}{F}$$
can be used to determine the image distance *l*’. The magnification *M* can be expressed as
(2)M=h'/ h=l'/l
where *h*′ is the image height, and *h* denotes the object height. 

The current demand for slim macro lenses in smartphones has led to the selection of COB packaging for the sensor in the fingerprint recognition optical system. The relevant sensor data are listed in Table 1. The reason for choosing these three wavelengths, 430 nm, 505 nm, and 580 nm, is that they coincide with the center wavelength of the red, green, and blue color filters built in the front glass of the liquid crystal module. Based on the imaging area of the sensor at 1.2 mm × 1.4 mm, its height corresponds to half of the diagonal length, which is *h*′ = 1.84 mm/2 = 0.92 mm. To prevent vignetting due to manufacturing inaccuracies, it is customary to set the height greater than half of the sensor’s diagonal length. Therefore, the image height is set to 1.0 mm. 

The fingerprint recognition system operates only in a specific area of the cell phone screen to lead that the object height *h* is set to 3.5 mm. Considering the COB packaging process requires that the image distance *l*′ is set to 0.88 mm. The object distance *l* is then obtained from Equation (2) as −3.08 mm. Using Equation (1) can determine the EFL of the lens to be *F* = 0.7 mm, which is a shorter EFL to result in a larger FOV. The half FOV *θ* can be approximated as *θ* ≈ *h*/*l* ≈ 65°. For lenses with finite conjugate distance, the working F-number (*F*/#*_w_*) [16] is commonly used for the lens design. It is expressed as
(3)$$F/{\#}_{w}\approx \left(1-M\right)\frac{EFL}{EPD}\;$$
where EPD is the entrance pupil diameter (EPD) of the lens. *F*/#*_w_* is a critical parameter in optical lens design that is closely related to both light flux and image resolution. A smaller *F*/#*_w_* value indicates a larger aperture, resulting in higher light flux and theoretically better image resolution. However, smaller *F*/#*_w_* values also pose greater challenges for lens design, necessitating a comprehensive consideration of various factors before setting the value to 2.8. The TTL of the lens refers to the distance between the first lens surface and the image plane within the optical system. After referring to the relevant patent literature [9], we have set the target value for the TTL of this lens to be less than 2.7 mm. 

The modulation transfer function (MTF) is used to evaluate the performance of optical systems and measure lens resolution. In this study, the image sensor is used as the imaging surface, and the cutoff frequency *f_cut_freq_* of the optical system is set by the pixel size of the sensor, that is,
(4)$$f_{cut\_freq}=\frac{1\;}{2\times pixel\;size}\;$$

The pixel size is 6.25 μm to result in *f_cut_freq_* = 80 lp/mm. The industry generally uses 1/2 cutoff frequency to evaluate the imaging quality in lens design. Therefore, we require MTF 40 lp/mm as the basis for the lens performance. Table 2 shows the target specifications for the lens in a fingerprint recognition system. 

According to Gaussian optics, a lens consisting of multiple lens elements with no afocal system can be equivalent to a single thin lens. Such a single lens often falls short in achieving optimal image resolution due to aberrations. Image quality is then improved through multiple lens elements for realizing aberration balance. The fingerprint recognition optical system requires an ultra-wide FOV, short TTL, and high RI. A three-lens-element architecture as shown in Figure 2a provides a solution for meeting the image quality requirements in fingerprint recognition [9]. These lens elements are sequentially negative, positive, and positive diopters to involve considering the following features: (a) lens 1 has a negative diopter, allowing the optical system to enhance the FOV and improve distortion; (b) lens 2 has a positive diopter, assisting lens 1 in improving the FOV, correcting distortion, and controlling the diameter of lens 1; (c) combining lens 3 with a positive diopter and a smaller focal length achieves focusing and controls the overall length of the lens; (d) the aperture is positioned between lens two and lens three, enabling an increase in aperture size and enhancement of image height; (e) this architecture effectively corrects Petzval [17]. 

The commonly used optical plastic materials include Cyclic Olefin Copolymer (COC), Cyclo Olefin Polymer (COP), and Polyester (OKP). Considering cost and manufacturability, this study utilizes COP and OKP. Lens 1 and lens 3 are selected to be made of COP material, with a refractive index of 1.53 and an Abbe number of 56.1. Lens 2 is paired with OKP material, which has a high refractive index of 1.64 and a low Abbe number of 22.4, aiming to reduce the chromatic aberration issues in the optical system [16]. Current lens designs are frequently derived from existing lenses. Lai [9] has proposed a layout shown in Figure 2a that serves as our initial lens structure, as it closely aligns with our requirements. The initial layout is then optimized through further process for achieving a better image quality. 

## 3. Lens Optimization for Fingerprint Recognition

Based on the target specifications, the ZEMAX merit function is used to define the optimization criteria, including EFL, TTL, lens central and edge thickness, air gap distance, and so on. During optimization, it is important to avoid interference between lenses, lens thickness that is too small, and excessive or insufficient aspheric curvature. These conditions can make lenses difficult or impossible to manufacture. After optimization, the image quality, including field curvature, distortion, RI, and MTF, should be evaluated to determine whether it meets the specifications. If it does not, the initial design may need to be revised or the specifications may need to be renegotiated. 

The initial design of the lens showed a sharp decline in MTF values in the outer field, as well as a significant difference between tangential and sagittal MTF values. To improve this, the optimization process adjusted the weight values of the lateral light aberration in each FOV. However, due to the large FOV of the lens, the RI in the outer field is lower than in the center. To improve the RI, the optimization process sacrificed some distortion control. This led to an increase in distortion at higher image heights. The optimized lens still has better performance than the initial design, but the distortion increase at high image heights is a potential limitation. 

In the optimization process, some parameters can be fixed, while others can be allowed to vary. The initial design of the lens in this paper uses three plastic aspheric lenses, which gives the system more freedom to optimize. However, due to cost and manufacturability concerns, the first and third lenses are made of COP material, while the second lens is made of OKP material. The optimization process begins by changing all curvature radii, lens thicknesses, and air gaps to variable. Once this is completed, the aspheric secondary surface constant and aspheric coefficient are gradually adjusted to variable. The optimization results show that the best performance is achieved by increasing the negative refractive index of the first lens, while shrinking the difference between the positive refractive indices of the second and third lenses. The aperture is also moved closer to the second lens. These changes result in a lens that meets the target specifications, including EFL, FOV, distortion, and MTF. 

The initial design of the lens was optimized using ZEMAX software (version 2023 R2) to achieve the desired specifications. The optimized layout is shown in Figure 2b. The optimized optical design parameters are shown in Table 3 and Table 4. The MTF diagram in Figure 3 shows that the MTF value at the center of the FOV is greater than 80%, and the MTF value at 0.7 FOV is greater than 70%. The field curvature diagram in Figure 4 shows that the maximum field curvature is around 0.1 mm. The distortion diagram shows that the distortion is negative distortion (barrel distortion), with a maximum value of around 8%, which is less than the target specification of 10%. This wide-angle lens exhibits a distortion of nearly 8%, but this aberration can be corrected using digital image correction techniques [12]. Additionally, the optical sensor receives reflected light from the finger in a periodic arrangement, leading to diffraction effects that reduce image resolution [18,19]. Building upon the methodologies outlined by Yang et. al. [18] and Feng et. al. [19], further enhancements can be made to mitigate these diffraction effects arising from the sensor’s periodic pixel arrangement, consequently improving image quality. 

Relative illumination is the ratio of the illuminance at the periphery of an image to the illuminance at the center. This study designed a wide-angle lens, which means that the brightness decreases more quickly as the angle increases and approaches the periphery. As shown in Figure 5, the RI is greater than 25% at an image height of 1.0 mm. The size and shape of the spot in the spot diagram can be used to evaluate the optical system aberration. For visual systems, the Airy disc diameter, which is the smallest possible spot size due to diffraction, is used as a reference. When using a CMOS sensor, the spot size is defined as twice the pixel size. Therefore, the spot size needs to be controlled at 2 × 6.25 μm = 12.5 μm. As shown in Figure 6, the spot sizes at the center, 0.7 FOV, and 1.0 FOV are all less than 12.5 μm. Most under-display fingerprint systems are currently applied to LCD and OLED displays, which have different light source spectrums [20]. Therefore, it is necessary to analyze the chromatic aberration of the lens. In this study, we used three design wavelengths for the lens: 430 nm, 505 nm, and 580 nm. These wavelengths can cover almost the entire light source distribution of LCD and OLED panels. As shown in Figure 7, the chromatic aberration distribution of the lens is approximately around 3 μm. The pixel size of the sensor we used is 6.25 μm. Therefore, the designed lens can be used for both LCD and OLED display panels. Figure 8 shows a fidelity of the fingerprint between the simulated image and the original image. As a result, the optimized lens meets the desired specifications. 

## 4. Lens Tolerance Analysis for Fingerprint Recognition 

Tolerance analysis is a critical step in the optical system design process. It is used to evaluate whether the system will meet optical performance requirements (such as MTF and other image quality criteria) and to minimize the cost of manufacturing, assembly, and calibration. By balancing optical performance and manufacturing cost, tolerance analysis can help to maximize the profitability of lens production. ZEMAX [15] provides a powerful and flexible tolerance analysis tool. It can be used to analyze a variety of tolerances, including curvature, thickness, position, refractive index, Abbe number, and aspheric coefficient. It can also be used to analyze the effects of eccentricity and tilt on lens surfaces and lens groups. 

This paper uses the sensitivity analysis function of the ZEMAX tolerance analysis tool to analyze the yield of a lens system with default tolerance settings. The results show that the yield of the system is sensitive to the tolerances of the lens surfaces and lens groups. By optimizing the tolerance settings, the yield of the system can be improved. According to the current industry processing and assembly capabilities, the table of tolerance settings for process yield analysis is shown in Table 5. Through sensitivity analysis (Sensitivity Analysis), the analysis was conducted using diffraction MTF average 40 lp/mm. The estimated MTF is 82.4% in the center field of view and 70.3% in the 0.7 FOV. The MTF analysis results are shown in Table 6. 

The study then used Monte Carlo analysis (Monte Carlo Analysis) to estimate the yield of the lens. Monte Carlo analysis is a method that simulates the overall impact of all tolerances on the lens at the same time. This simulation method generates a series of random lenses that meet the specified tolerance setting values, and then evaluates the lenses using the evaluation criteria. According to Table 5, the number of tolerance operations in this study was set to 56. In order to meet the appropriate sampling analysis, the number of Monte Carlo simulations should be at least 3200 times. The analysis results are shown in Table 7. When the yield is greater than 80%, the MTF is 82.5% in the center field of view and 70.2% in the 0.7 field, both of which meet the mass production yield requirements. 

## 5. Conclusions 

This work presents a wide-angle lens design for fingerprint recognition systems, pushing the boundaries of FOV and compactness. Boasting a remarkable 126° FOV within a miniscule 2.54 mm total length, the lens empowers capturing more comprehensive fingerprint regions even in space-constrained scenarios like mobile devices. Furthermore, it guarantees consistent high-quality image acquisition through a minimum edge RI exceeding 25%, significantly surpassing existing solutions. Crucially, tolerance analysis confirms the feasibility of mass production with a 98% yield within current manufacturing tolerances, paving the way for widespread adoption in fingerprint recognition systems. As a result, this lens design represents a significant leap forward in fingerprint capture capabilities, promising enhanced accuracy and broader applicability in diverse fingerprint recognition applications. 

## Figures and Tables

**Figure 1 micromachines-15-00386-f001:**
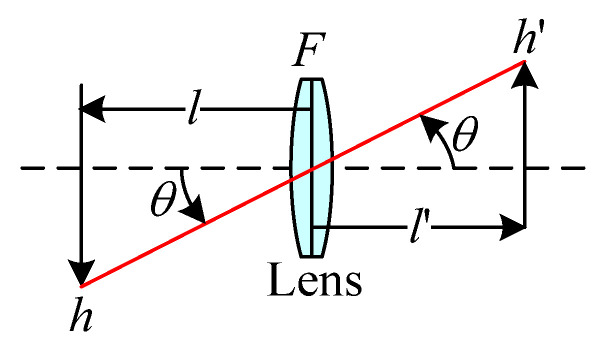
Image lens system.

**Figure 2 micromachines-15-00386-f002:**
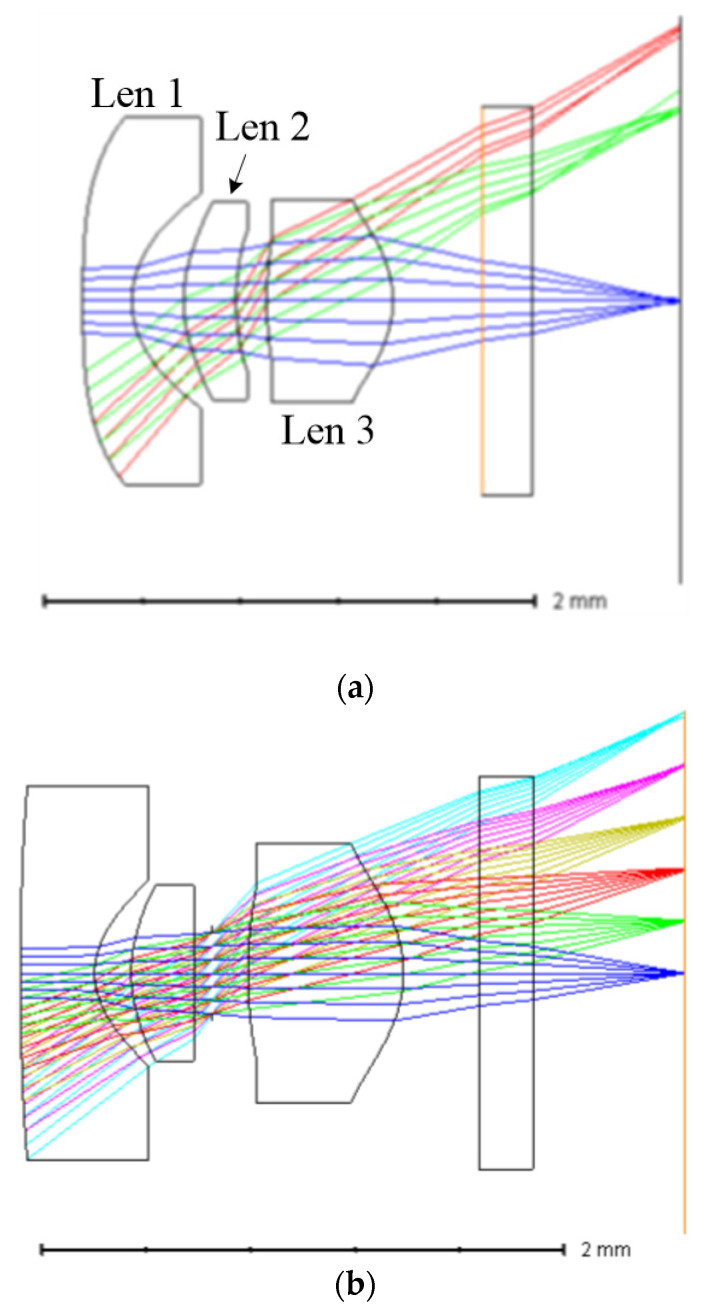
The lens layout. (**a**) The initial lens layout [9]. (**b**) The optimization lens layout.

**Figure 3 micromachines-15-00386-f003:**
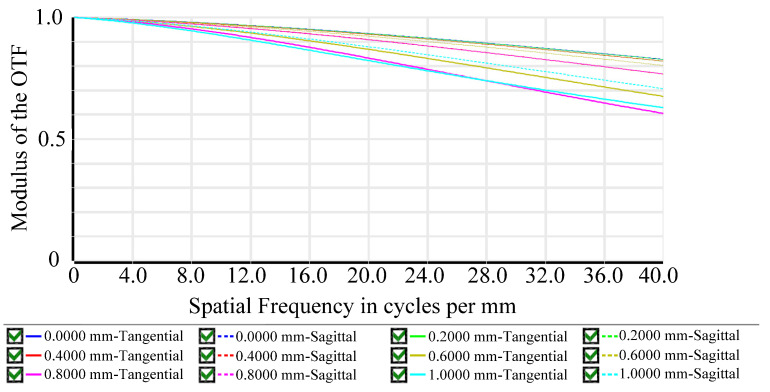
MTF diagram.

**Figure 4 micromachines-15-00386-f004:**
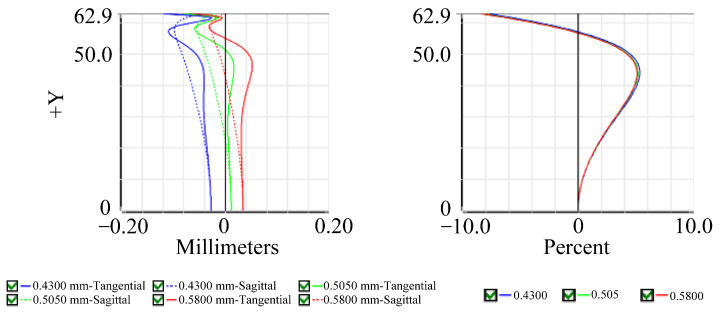
Field curvature and distortion diagram.

**Figure 5 micromachines-15-00386-f005:**
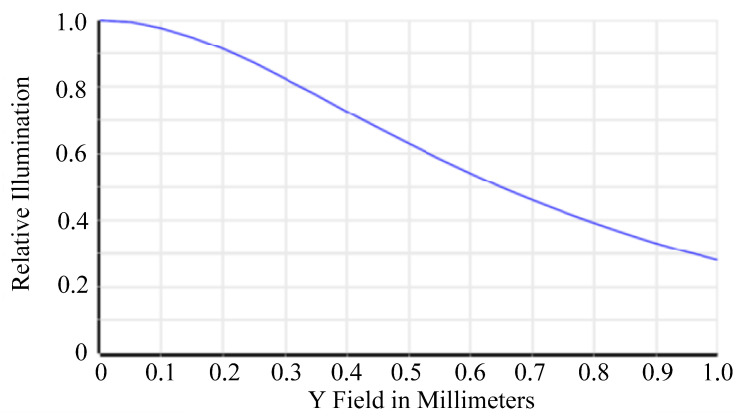
Relative illumination diagram.

**Figure 6 micromachines-15-00386-f006:**
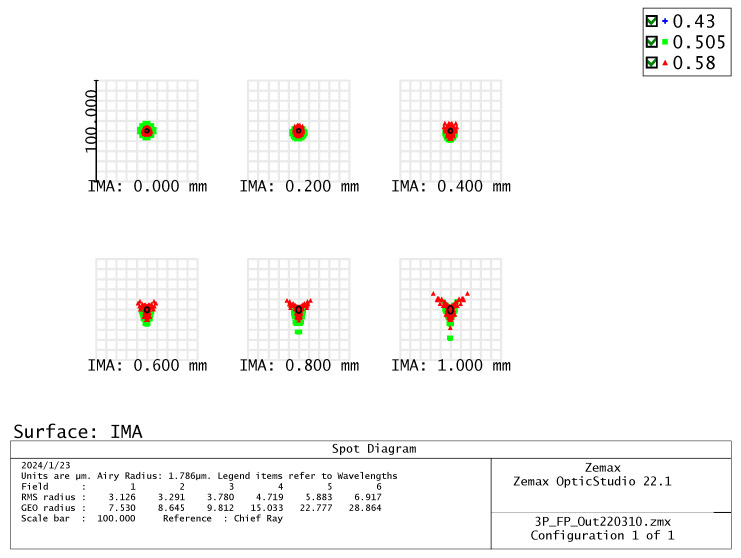
Spot diagram.

**Figure 7 micromachines-15-00386-f007:**
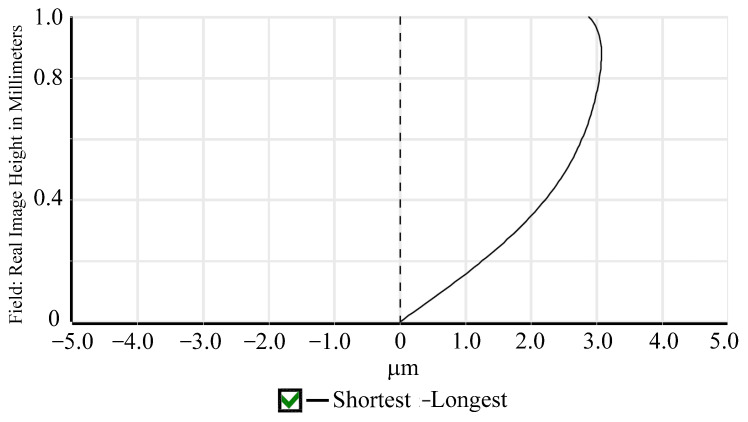
Lateral color aberration.

**Figure 8 micromachines-15-00386-f008:**
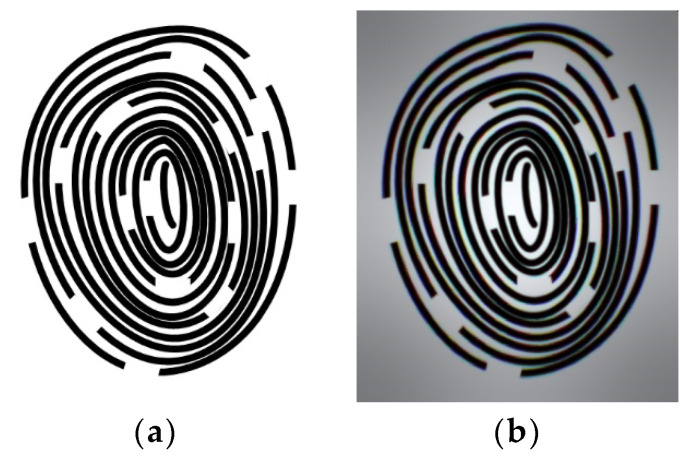
Image simulation results: (**a**) original photo, (**b**) simulation image.

**Table 1 micromachines-15-00386-t001:** Sensor specification.

Item	Specification
Resolution	192 × 224
Pixel size	6.25 μm × 6.25 μm
Image area	1.2 mm × 1.4 mm
Diagonal length	1.84 mm
Wavelengths	430 nm (B); 505 nm (G); 580 nm (R)

**Table 2 micromachines-15-00386-t002:** Lens specification.

Items		Specification
EFL (Effect Focal Length)		<0.7 mm
*F*/#*_w_*		2.8 ± 5%
FOV (Field of View)		around 130°
Optical distortion		<10% (@IH = 1.0)
RI (Relative Illumination)		>25% (@IH = 1.0)
TTL (Total Track Length)		<2.7 mm
Image height (IH)		1.0 mm
MTF	On Axis	>80% @40 lp/mm
	70% Field	>70% @40 lp/mm

**Table 3 micromachines-15-00386-t003:** The optimization lens parameters.

Surface	Type	Radius	Thickness	Material
OBJ	STANDARD	Infinity	0.9	-
1	STANDARD	Infinity	1.33	BK7
2	STANDARD	Infinity	0.12	-
3	EVENASPH	14.681	0.283	COP
4	EVENASPH	0.277	0.14	-
5	EVENASPH	0.554	0.239	OKP
6	EVENASPH	−60.386	0.07	-
STO	STANDARD	Infinity	0.137	-
8	EVENASPH	1.722	0.596	COP
9	EVENASPH	−0.469	0.287	-
10	STANDARD	Infinity	0.21	BK7
11	STANDARD	Infinity	0.58	-
IMA	STANDARD	Infinity	0	-

**Table 4 micromachines-15-00386-t004:** The optimization aspherical coefficient.

Aspherical Coefficient	Surface					
	3	4	5	6	8	9
K	−21.648	−1.197	−1.766	−11.061	−56.293	−1.034
A2	0	0	0	0	0	0
A4	0.048	−0.0506	−0.5293	0.0626	1.1571	0.1925
A6	−0.0212	−3.2071	−1.1382	14.563	−1.4885	2.4377
A8	−0.0323	27.43	16.814	−28.037	−28.048	6.4733
A10	0.0207	36.77	203.06	497.32	32.088	2.9455
A12	0	−540.26	800.54	−563.39	733.97	−54.65
A14	0	−4593.5	−4629.4	−265,788	878.49	−116.27
A16	0	−31,867	−88,073	2,074,393	−22,386	213.55

**Table 5 micromachines-15-00386-t005:** Tolerance analysis setting.

Items		Setting
Surface	Radius (mm)	±0.01
	Thickness (mm)	±0.003
	Decenter X (mm)	±0.002
	Decenter Y (mm)	±0.002
	Tilt X (°)	±0.02
	Tilt Y (°)	±0.02
Element	Decenter X (mm)	±0.002
	Decenter Y (mm)	±0.002
	Tilt X (°)	±0.1
	Tilt Y (°)	±0.1
Index	Refractive index	±0.0005
	Abbe number	±0.8%

**Table 6 micromachines-15-00386-t006:** MTF tolerance analysis results.

Item	Central FOV	0.7 Field
Nominal MTF	82.70%	72.70%
Estimated change	−0.30%	−2.40%
Estimated MTF	82.40%	70.30%

**Table 7 micromachines-15-00386-t007:** Yield rate simulation results by Monte Carlo.

Item	Central FOV	0.7 Field
98% >	0.823 @40 lp/mm	0.679 @40 lp/mm
90% >	0.824 @40 lp/mm	0.695 @40 lp/mm
80% >	0.825 @40 lp/mm	0.702 @40 lp/mm
50% >	0.826 @40 lp/mm	0.713 @40 lp/mm
20% >	0.827 @40 lp/mm	0.720 @40 lp/mm
10% >	0.828 @40 lp/mm	0.723 @40 lp/mm
2% >	0.829 @40 lp/mm	0.726 @40 lp/mm

## Data Availability

Data will be made available on request.

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
