# Peer review of "Wide Field of View Under-Panel Optical Lens Design for Fingerprint Recognition of Smartphone"

_micromachines, 2024, doi:10.3390/mi15030386_

Round 1

Reviewer 1 Report

Comments and Suggestions for Authors

Please see attached review report.

Reviewer 2 Report

Comments and Suggestions for Authors

The authors have provided a work on the optical lens design for fingerprint recognition of LED smartphone, which sounds quite interesting. Before acceptance, some issues in the manuscript need to be addressed, as follows:

(1)    The introduction seems too brief, more discussions on the design on the wide-angle lenses should be done.

(2)    In equation (1), the effective focal length is F, not F’, please correct it.

(3)    In Table 1, for three wavelengths, it is better to label the “R, G, B” for each value, such as 430 nm (B)…

(4)    In Table 1, diagonal length should be Diagonal length

(5)    In Figure 2, each optical lens should be given a name

(6)    In equation (3), why the value of F/#w is set to 2.8?

(7)    Page 5, is COMS sensor is wrong?

(8)    What is the novelty of this work compared with others? And there is not any data supporting the title of LED smartphone.

Round 2

Reviewer 2 Report

Comments and Suggestions for Authors

Most have been addressed except for

1. Some figures still do not have good clarity, such as FIGS. 4, 5, 6, 8

2. The font inconsistency remains , such as TAB 2

3. Figure 2 and 3 can be combined to a figure, labeled as (a) and (b)

Author Response

Comments and Suggestions for Authors

Most have been addressed except for

  1. Some figures still do not have good clarity, such as FIGS. 4, 5, 6, 8.

Response: Thanks for reviewer’s suggestion.

We have tried to make the Figures clear more, as shown in Figure 3,4, 5, and 7.

  1. The font inconsistency remains, such as TAB 2.

Response: Thanks for reviewer’s suggestion.

We have made the font of all Tables to be the ‘Palatino Linotype’ that is the template font for Micromachines.

  1. Figure 2 and 3 can be combined to a figure, labeled as (a) and (b)

Response: Thanks for reviewer’s suggestion.

We have combined Figure 2 and 3 to be Figure 2.